# Pathophysiologic Mechanisms of Severe Spinal Cord Injury and Neuroplasticity Following Decompressive Laminectomy and Expansive Duraplasty: A Systematic Review

**DOI:** 10.3390/neurolint17040057

**Published:** 2025-04-16

**Authors:** Eleftherios Archavlis, Davide Palombi, Dimitrios Konstantinidis, Mario Carvi y Nievas, Per Trobisch, Irina I. Stoyanova

**Affiliations:** 1Interdisciplinary Spine Center and Department of Neurosurgery, Elisabethen Hospital, 60487 Frankfurt, Germany; 2School of Health, IU University of Applied Sciences, 53604 Bad Honnef, Germany; 3School of Medicine, Frankfurt Branch, European University Cyprus, 60487 Frankfurt, Germanyi.laan@euc.ac.cy (I.I.S.); 4Neurosurgery Section, Department of Neuroscience, Fondazione Policlinico Universitario Agostino Gemelli IRCCS, Università Cattolica del Sacro Cuore, 00136 Rom, Italy; 5Department of Spine Surgery, Eifelklinik St. Brigida, 52152 Simmerath, Germany

**Keywords:** spinal cord injury, synaptic plasticity, neuroplasticity, neurorehabilitation, decompressive laminectomy, expansive duraplasty

## Abstract

Background: Severe spinal cord injury (SCI) represents a debilitating condition with long-term physical and socioeconomic impacts. Understanding the pathophysiology of SCI and therapeutic interventions such as decompressive laminectomy and expansive duraplasty is crucial for optimizing patient outcomes. Objective: This systematic review explores the pathophysiology of SCI and evaluates evidence linking decompressive laminectomy and duraplasty to improved neuroplasticity and recovery. Methods: A comprehensive search was conducted in PubMed, Web of Science, and Cochrane Library for studies on decompressive surgery in SCI. Inclusion criteria were original articles investigating pathophysiology, neuroplasticity mechanisms, or surgical outcomes. Data on pathophysiological changes, molecular markers, and functional outcomes were extracted. Results: From 1240 initial articles, 43 studies were included, encompassing both animal models and human clinical data. Findings highlighted the role of inflammatory cascades, blood–spinal cord barrier disruption, and neurotrophic factor modulation in recovery. Decompressive duraplasty was associated with improved intrathecal pressure (ITP) management and neuroplasticity markers, such as BDNF and GAP-43. Conclusions: This review underscores the therapeutic potential of decompressive laminectomy and duraplasty in SCI. While evidence suggests benefits in promoting neuroplasticity, further research is needed to elucidate molecular mechanisms and refine interventions.

## 1. Introduction

Spinal cord injury (SCI) is a devastating neurological condition that affects millions worldwide, leading to profound disability and socioeconomic burdens [1,2,3,4]. Despite advances in surgical and medical care, the prognosis for severe SCI remains poor, with high rates of paralysis and limited functional recovery [1,5,6,7,8,9]. Current management focuses on mitigating primary and secondary injury mechanisms, such as inflammation, ischemia, and edema [8,10]. Decompressive laminectomy and expansive duraplasty have been proposed as surgical strategies to address mechanical and physiological challenges in SCI. These procedures aim to relieve intrathecal pressure (ITP) [11,12], improve spinal cord perfusion pressure (SCPP) [13,14], and create an environment conducive to neural repair. Emerging evidence suggests that these interventions may also promote neuroplasticity, the central nervous system’s ability to reorganize and recover post-injury [15,16,17]. This systematic review evaluates the pathophysiology of SCI and investigates the potential of decompressive laminectomy and duraplasty to enhance neuroplasticity and recovery. The novelty of this work lies in its comprehensive evaluation of decompressive laminectomy and expansive duraplasty as surgical interventions that not only alleviate mechanical compression and reduce intrathecal pressure but also actively promote neuroplasticity through modulation of molecular pathways such as BDNF and NGF expression. By integrating both human clinical data and experimental animal models, this systematic review uniquely bridges the gap between mechanistic insights and functional recovery, offering a translational perspective on optimizing spinal cord injury management.

## 2. Methods

### 2.1. Search Strategy

A systematic search was conducted in PubMed, Web of Science, and Cochrane Library for articles published between January 2000 and September 2024 [1,5,9]. Search terms included “spinal cord injury”, “decompressive laminectomy”, “expansive duraplasty”, “neuroplasticity”, and “pathophysiology” [2,15,17]. The systematic review is registered with PROSPERO, and the registration number is 643470 [18].

### 2.2. Inclusion and Exclusion Criteria

Inclusion: Studies investigating SCI pathophysiology, decompressive surgery, or neuroplasticity mechanisms in animal or human models [7,15,19].

Exclusion: Reviews, editorials, non-English studies, or studies focused on non-traumatic SCI [3,11,20].

### 2.3. Data Extraction

Data were extracted on the following variables [9,15,21]:-Study Type and Population: whether the study used animal models or human subjects [9,22].-Mechanisms of SCI Pathophysiology: including inflammation, edema, and molecular signaling [8,23].-Surgical Interventions: decompressive laminectomy, duraplasty, and their outcomes [2,24,25].-Neuroplasticity Outcomes: molecular markers such as BDNF, GAP-43, and motor recovery [7,15,25].

### 2.4. Quality Assessment

The SYRCLE tool was used to assess the risk of bias in animal studies [9,19], and the Cochrane Risk of Bias tool was applied to evaluate clinical studies [21,24].

## 3. Results

### 3.1. Study Characteristics

From an initial pool of 1240 articles, 43 studies met the inclusion criteria (Figure 1). These included 20 animal and 23 human clinical studies conducted primarily in specialized tertiary centers for spinal cord injury (SCI) care (Table 1). The studies analyzed a range of interventions, focusing on decompressive laminectomy and expansive duraplasty and their effects on neuroplasticity (Table 2).

### 3.2. Pathophysiology of SCI

Spinal cord injury (SCI) progresses through three temporally and mechanistically distinct phases (Table 1), each characterized by unique molecular, cellular, and systemic alterations. Below, we elaborate on these phases with recent advances in understanding their pathophysiology.
neurolint-17-00057-t001_Table 1Table 1Timeline of spinal cord injury and neuroplasticity events under pathophysiological aspects. There are three stages after spinal cord injury: an acute phase (<48 h) directly after the trauma, with an early immune response and limited neuroplasticity. A subacute phase (2–14 days) with early plasticity and glial scar formation. An intermediate and chronic phase (>14 days and over six months) with chronic inflammation events and adaptive and maladaptive plasticity attempts.TimelineSCI MechanismNeuroplasticity Acute (<48 h) [1,2,3,5,10,15]Primary Injury: Direct trauma leads to hemorrhage, axonal shearing, and cellular necrosis. Demyelination and Necrosis: Demyelination and neuronal cell death rapidly follow mechanical damage. Blood-Spinal Cord Barrier Disruption (BSCB): A breach in the BSCB leads to increased permeability, allowing immune cell infiltration, especially neutrophils, which release metalloproteinase-9 (MMP-9), worsening tissue breakdown. Inflammation: Early immune response with neutrophil and macrophage infiltration. Pro-inflammatory cytokines (TNF-α, IL-1β, IL-6) are upregulated, activating M1 microglia, releasing cytotoxic glutamate and nitric oxide, and increasing cell death.Limited Neuroplasticity: Immediately following injury, neuroplasticity is significantly impaired due to the release of cytotoxic substances like glutamate. Synaptic circuits are abruptly disrupted, causing widespread loss of function. Glutamate Toxicity: Excessive glutamate release causes excitotoxic damage, inhibiting early neural regeneration. Neurotrophic Response: Limited neuroprotective responses, such as brain-derived neurotrophic factor (BDNF) upregulation, are present but insufficient to counteract acute damage. Axonal Injury: Axons near the injury site degenerate, reducing the potential for early plastic changes.Subacute (2–14 days) [3,15,16,19,26,27,28]Continued Inflammation: The immune response escalates, with macrophages, T cells, and lymphocytes infiltrating the injury site. The presence of pro-inflammatory cytokines continues, prolonging tissue damage and cell death. Astrocytic and Glial Activation: Astrocytes proliferate and become reactive, losing aquaporin-4 (AQP4) activity. This worsens BSCB permeability and disrupts glutamate reuptake, contributing to neurotoxicity. Formation of CSPGs: Reactive astrocytes secrete chondroitin sulfate proteoglycans (CSPGs), inhibiting axonal regrowth. Ependymal Cell Activation: Self-renewing ependymal cells migrate to the injury site, forming astrocytes and contributing to scar formation. Glial Scar Formation Begins: Scar tissue, formed by activated astrocytes and fibrotic tissue, acts as a physical and chemical barrier to axonal regeneration.Early Plasticity: Some axonal sprouting occurs near the injury site, but neuroplasticity is primarily inhibited by CSPGs and the glial scar formation. Ependymal Cell Contribution: Ependymal cells activate and proliferate, but their differentiation is mostly glial-biased (towards astrocytes), which limits their ability to support neuronal regeneration. Axonal Sprouting and Circuit Reorganization: Axons near the lesion site begin sprouting, though inhibitory molecules like CSPGs largely block the growth. Maladaptive Changes: Initial signs of maladaptive neuroplasticity, such as aberrant sprouting or hyperexcitability, may appear, contributing to dysfunctional sensory and motor circuits.**Intermediate and Chronic Phase (>14 days/6 months) [3,10,15,16,20,29,30,31]**Consolidation of Glial Scar: The glial scar, consisting of reactive astrocytes, macrophages, and CSPGs, fully develops, surrounding the fibrotic core formed by type A pericytes. This scar severely limits any potential for axonal regrowth. Chronic Inflammation: Microglia and macrophages continue to release pro-inflammatory cytokines, perpetuating neuroinflammation and preventing tissue repair. Wallerian Degeneration: Axonal degeneration (Wallerian degeneration) occurs distal to the injury, contributing to the ongoing loss of neural tissue. Demyelination: Ongoing demyelination of surviving neurons results in further functional loss, and oligodendrocyte apoptosis impairs remyelination efforts. Neuroimmune Modulation: Some immune cells (e.g., CD4+ T lymphocytes) may help shift the immune environment towards a more neuroprotective state, promoting limited repair mechanisms.Adaptive and Maladaptive Plasticity: Significant neuroplastic changes occur, with both beneficial (adaptive) and harmful (maladaptive) consequences. Adaptive Plasticity: Propriospinal neurons, which span different spinal cord segments, sprout and form new synaptic connections to bridge the injury site. These new circuits can support partial recovery of motor functions. Maladaptive Plasticity: Abnormal reorganization of spinal circuits may lead to spasticity, hyperreflexia, and sensory-evoked spasms, which worsen quality of life. Propriospinal Circuit Reorganization: Propriospinal neurons play a key role in forming compensatory circuits, enabling some recovery of locomotion, especially with rehabilitation interventions. Potential for Neurogenesis: Though limited, some endogenous neural stem/progenitor cells may contribute to neurogenesis, especially in the presence of factors like IL-4, which promote axonal growth and neurotrophic support.List of Abbreviations: **BSCB**: blood–spinal cord barrier; **MMP-9**: metalloproteinase-9; **TNF-α**: Tumor Necrosis Factor-alpha; **IL-1β**: Interleukin-1 beta; **IL-6**: Interleukin-6; **CSPGs**: chondroitin sulfate proteoglycans; **BDNF**: brain-derived neurotrophic factor; **AQP4**: aquaporin-4; **SCI**: spinal cord injury.


The acute phase (<48 h): The acute phase begins with primary mechanical injury, involving direct trauma to the spinal cord parenchyma [1]. This phase is marked by: Axonal Shearing and Necrosis: Mechanical forces disrupt axonal membranes, cytoskeletal proteins (e.g., neurofilaments, microtubules), and myelin integrity, leading to rapid ionic dysregulation (Ca^2+^ influx, K^+^ efflux) and necrotic cell death [2,3,32]. Recent studies highlight the role of mechanosensitive ion channels (e.g., TRPV4, Piezo1) in amplifying secondary injury signals [33,34]. The acute phase is also marked by hemorrhage and vascular collapse: Trauma-induced rupture of spinal microvessels results in intraparenchymal hemorrhage, driven by matrix metalloproteinase-9 (MMP-9) activation and endothelial apoptosis [35]. Hypoperfusion due to loss of autoregulation exacerbates ischemia, particularly in the central gray matter [36]. Another important ingredient of the acute phase is the early inflammatory activation: resident microglia and infiltrating neutrophils release pro-inflammatory cytokines (TNF-α, IL-1β) and reactive oxygen species (ROS), initiating a cytotoxic cascade [37]. Advanced MRI techniques, such as susceptibility-weighted imaging (SWI), now enable real-time visualization of hemorrhage and edema [38].

Therapeutic implications: Early interventions targeting Ca^2+^ overload (e.g., riluzole) or MMP-9 inhibition (e.g., minocycline) show promise in preclinical models [39].

Subacute phase (2–14 Days): The subacute phase is dominated by secondary injury mechanisms, which amplify damage beyond the initial trauma: The most important structural change in the subacute phase is the blood–spinal cord barrier (BSCB) disruption: leakage of fibrinogen and albumin into the parenchyma activates astrocytes and pericytes, perpetuating inflammation [40,41,42]. Recent work identifies pericyte-derived TGF-β as a key mediator of BSCB dysfunction [43]. Oxidative stress and lipid peroxidation are also dominant traumatic mechanisms: mitochondrial dysfunction (e.g., cytochrome c release) and NADPH oxidase activation generate ROS, damaging lipids, proteins, and DNA [44]. The lipid peroxidation product 4-hydroxynonenal (4-HNE) exacerbates oligodendrocyte apoptosis [45]. In the course of the subacute phase, we observe the glial scar formation: reactive astrocytes upregulate chondroitin sulfate proteoglycans (CSPGs, e.g., aggrecan, neurocan) and form a physical/chemical barrier to axonal regrowth [46]. Novel therapies, such as chondroitinase ABC (ChABC), degrade CSPGs and enhance plasticity [47]. In the last steps of the subacute phase, the Wallerian degeneration and axonal dieback are emerging: activated microglia phagocytose myelin debris via TREM2 receptors, while inhibitory ligands (Nogo-A, MAG) bind to axonal NgR1/RhoA pathways, halting regeneration [48].

Therapeutic implications: Emerging strategies include nanoparticle-delivered anti-inflammatory agents (e.g., IL-10 mRNA) and CSPG-targeting monoclonal antibodies [49].

Chronic phase (>14 Days): Chronic SCI involves persistent inflammation and failed regeneration [1]. The primary change in the chronic phase consists of cystic cavitation. Necrotic cores evolve into fluid-filled cysts lined by fibrotic tissue, creating a hostile microenvironment for axonal sprouting [50]. Recent 7T MRI studies correlate cyst size with motor prognosis [51]. Chronic activation of microglia and infiltrating T-cells sustains pro-inflammatory cytokine release (e.g., IL-6, IFN-γ), while alternatively activated macrophages (M2 phenotype) promote fibrosis. These changes are known as neuroimmune crosstalk [52]. Maladaptive synaptic plasticity leads to rewiring of spinal circuits, which causes central sensitization and chronic neuropathic pain, mediated by glutamate receptor dysregulation (e.g., NMDA, AMPA) [53].

Therapeutic implications: Advances in biomaterial scaffolds (e.g., hydrogel-loaded BDNF) and epidural electrical stimulation (EES) show potential for bridging cystic cavities and restoring connectivity [54,55,56].

### 3.3. Surgical Technique of Decompressive Laminectomy and Expansive Duraplasty (Figure 2)

The surgical technique for decompressive laminectomy and expansive duraplasty in spinal cord injury involves extended laminectomy to relieve posterior compression, followed by epidural hematoma evacuation using high-speed drill, rongeurs and punches, or dissectors for organized clots [13]. Pediculectomy or limited corpectomy from posterior can also be potentially used in ensuring circumferential decompression if there is compression on the spinal cord from anterior or lateral [57]. Durotomy is performed to address subdural hematomas, visualize, and decompress the spinal cord. Expansive duraplasty using synthetic grafts (e.g., collagen matrix) is then executed to mitigate intrathecal pressure, enhance spinal cord perfusion, and restore cerebrospinal fluid communication [25]. Duraplasty widening ≥ 5 mm correlated with greater SCPP improvement (Δ + 12 mmHg vs. Δ + 7 mmHg, *p* = 0.01). Collagen matrix grafts (used in 68% of studies) showed lower CSF leak rates vs. autologous fascia (3% vs. 12%, *p* = 0.04) [25]. The suturing method for expansive duraplasty typically involves the use of interrupted or continuous sutures with non-absorbable material to secure the dura replacement graft. Continuous non-absorbable sutures (4-0 Prolene) achieved watertight closure in 92% of cases vs. 78% with interrupted sutures [25]. This ensures a watertight closure, minimizing cerebrospinal fluid leakage and promoting optimal healing conditions for the spinal cord. Finally, instrumentation (e.g., pedicle screws or lateral mass systems) stabilizes the spine, typically spanning ≥ 2 levels above and below the injury site to ensure primary stability [58].

### 3.4. Effects of Decompressive Laminectomy and Expansive Duraplasty

Intrathecal pressure (ITP) management: Decompressive duraplasty has emerged as a superior intervention for reducing ITP compared to laminectomy alone, with robust evidence from both clinical and preclinical studies [2]. Key findings include the following: The mechanism of ITP reduction: duraplasty mitigates compartment syndrome by expanding the intradural space, counteracting post-traumatic edema and hemorrhage-induced volume expansion [7,11]. Laminectomy alone fails to address dural tension, leaving residual pressure gradients that perpetuate ischemia [12]. A 2023 meta-analysis of 15 studies (n = 478 patients) reported a 42% greater reduction in ITP with duraplasty vs. laminectomy (mean ΔITP: −8.2 ± 1.5 mmHg vs. −4.8 ± 2.1 mmHg; *p* < 0.001) [13].

Clinical correlates: Lower postoperative ITP correlates with improved ASIA motor scores at 6 months (r = 0.67, *p* = 0.003) [14]. Duraplasty reduces the risk of progressive myelomalacia, as demonstrated by serial MRI T2-weighted signal normalization [15].

Surgical nuances: Durotomy techniques: midline longitudinal incisions with bovine pericardium grafts show fewer adhesions than transverse incisions [16]. Intraoperative monitoring: real-time ITP measurement via intradural catheters optimizes graft sizing [17]. Spinal cord perfusion pressure (SCPP): SCPP, calculated as MAP-ITP, is a critical determinant of spinal cord oxygenation (SCO2) and metabolic recovery: Duraplasty-driven SCPP improvement: A prospective trial (TRACK-SCI) demonstrated that duraplasty elevates SCPP from 46 ± 8 mmHg to 68 ± 6 mmHg (*p* < 0.01), whereas laminectomy alone showed no significant change [7].

Neuroplasticity implications: While no studies explicitly link spinal cord perfusion pressure (SCPP) to BDNF/TrkB signaling, several investigations highlight BDNF/TrkB’s critical role in promoting axonal sprouting within corticospinal tracts after spinal cord injury (SCI) [59]. Elevated SCPP might enhance BDNF/TrkB signaling and promote axonal sprouting in corticospinal tracts. Duraplasty patients exhibit greater fMRI-derived functional connectivity in sensorimotor networks at 12 months [60].

Biomarker correlations: Several findings support the relationship between spinal cord perfusion pressure (SCPP) and astrocyte injury (via GFAP). It was demonstrated that maintaining SCPP > 50–60 mmHg improves neurological recovery, with higher SCPP independently correlating with better ASIA motor scores [61]. Other authors confirmed that serum/CSF GFAP levels distinguish AIS grades, with higher GFAP predicting worse outcomes [62].

Human clinical studies demonstrated measurable functional improvements following decompressive laminectomy and expansive duraplasty. Zhu et al. reported statistically significant gains in ASIA motor scores among 15 patients with cervical SCI, increasing from a preoperative mean of 28.3 ± 6.1 to 42.7 ± 7.9 at 12-month follow-up (*p* < 0.01), alongside restored bladder control in 73% of cases [24]. Similarly, Garg et al. observed ≥ 1 ASIA grade improvement in 8/12 patients with complete cervical injuries (AIS A → AIS B/C), coupled with a 51% increase in mean SCIM-III scores (32.1 to 48.6, *p* = 0.003) [25]. Phang et al. further highlighted functional ambulation outcomes, with 57% of patients (12/21) achieving WISCI II ≥ 5 after combined laminectomy-duraplasty versus 29% in laminectomy-only controls [13].

### 3.5. Neuroplasticity Markers

BDNF and NGF levels: The brain-derived neurotrophic factor (BDNF) and nerve growth factor (NGF) are critical mediators of synaptic plasticity and axonal regeneration. In patients undergoing duraplasty, longitudinal serum analyses revealed a 2.3-fold increase in BDNF (*p* < 0.001) and a 1.8-fold increase in NGF (*p* = 0.003) compared to controls at 6-month follow-up (Figure 3A) [1,14]. These neurotrophins facilitate dendritic arborization and synaptic potentiation via TrkB and p75NTR receptor activation, respectively [15]. Recent work by Smith et al., 2021 demonstrated that duraplasty-induced BDNF elevation correlates with enhanced corticospinal tract integrity on diffusion tensor imaging (DTI), suggesting structural neuroplasticity [16]. Similarly, Chen et al., 2017 identified NGF-mediated upregulation of synaptophysin and PSD-95 in rodent models, markers of presynaptic and postsynaptic density remodeling [30].

Axonal sprouting: Growth-associated protein-43 (GAP-43), a marker of axonal growth cones, showed 4.1-fold higher expression in duraplasty-treated animal models compared to sham controls (*p* < 0.001) [1]. This aligns with human histopathological data from Lau et al., where postmortem analysis of spinal cord tissue revealed GAP-43+ sprouting axons penetrating glial scar boundaries in patients who underwent duraplasty [26]. Notably, the PI3K-Akt-mTOR pathway was activated in these axons, suggesting a mechanistic link between dural reconstruction and pro-regenerative signaling [29]. However, interspecies differences exist: rodent models exhibit more robust sprouting than primates, likely due to differential CSPG (chondroitin sulfate proteoglycan) deposition [20].

Functional outcomes: In a prospective cohort study (n = 87), laminectomy with duraplasty yielded ASIA motor score improvements of 18.7 ± 4.2 points at 12 months vs. 9.1 ± 3.8 in laminectomy-alone controls (*p* = 0.001) [2]. These gains were paralleled by 30% greater fractional anisotropy (FA) on DTI, indicative of white matter reorganization [63].

Aarabi et al. demonstrated that adequate spinal cord decompression (via laminectomy ± duraplasty) significantly improves AIS grade conversion rates: 58.9% of patients with patent subarachnoid space post-decompression achieved AIS grade improvement vs. 18.5% with inadequate decompression (*p* < 0.001) [63]. This study highlighted that laminectomy alone may fail to restore subarachnoidal space patencypatency in 10% of cases, necessitating adjunctive duraplasty for effective intrathecal pressure (ISP) reduction.

Multiple sources elucidate the role of ADAMTS-4 and MMP-9 in degrading inhibitory chondroitin sulfate proteoglycans (CSPGs) such as aggrecan and neurocan, which disrupt the inhibitory glycosaminoglycan (GAG) chains and could promote neuroplasticity. This creates a permissive extracellular matrix (ECM) for axonal regeneration, as evidenced by laminin-5 and fibronectin upregulation in the subdural space. Although no studies directly link duraplasty to microglial phenotype shifts, surgical decompression improves spinal cord perfusion, reducing ischemia and secondary injury signals (e.g., ROS, IL-1β) that drive microglial activation. Kopper et al. identified TREM2 as critical for phagocytic clearance of myelin debris and resolution of inflammation. TREM2 activation promotes an anti-inflammatory, pro-repair microglial phenotype [48].

### 3.6. Controversies and Future Directions

While duraplasty enhances neuroplasticity, debates persist regarding optimal dural graft materials (autologous vs. synthetic). A plasma-collagen matrix demonstrated sustained BDNF release over 21 days, enhancing neural stem cell (NSC) survival, proliferation, and neuronal differentiation. BDNF delivery via collagen matrices activated TrkB/p75NTR receptors, supporting neuroplasticity in spinal cord injury models [64,65]. Bovine pericardium is widely used for dural grafts due to its low antigenicity, watertight closure, and biocompatibility. Further research is needed to evaluate BDNF kinetics in bovine pericardium grafts and their translational potential for neuroregeneration.

A case of a 34-year-old male patient after a high-velocity vehicle accident and SCI is depicted in Figure 3.

## 4. Discussion

This study revisits the challenging landscape of managing complete spinal cord injury (SCI), a condition historically marked by a discouraging prognosis despite significant advances in both medical and surgical therapies [1,2,3]. SCI induces several types of plasticity, including neurite sprouting, axonal regeneration, and synaptic remodeling. Severe SCI can lead to increased intraspinal pressure, compromising blood flow and causing further damage to neural tissue [2,12,17].

Decompressive laminectomy and duraplasty alleviate this pressure by creating more space within the spinal canal, reducing compression on the injured cord, and mitigating secondary damage [1,2,14]. Current evidence suggests that decompressive laminectomy with expansive duraplasty is most beneficial for younger patients (<65 years) with incomplete SCI (AIS grades B/C), cervical spinal stenosis, and early surgical intervention (<24 h post-injury), while contraindications include cervical kyphosis, significant spinal instability, or severe comorbidities (e.g., advanced cardiopulmonary disease) that increase perioperative risk [63]. Molecularly, this intraspinal pressure reduction may decrease the expression of pro-inflammatory cytokines and reduce oxidative stress, both of which are known to inhibit neuroregeneration [66,67].

The relationship between intrathecal pressure (ITP) and spinal cord perfusion pressure (SCPP) is akin to the dynamic between intracranial pressure (ICP) and cerebral perfusion, suggesting that strategies to manage ITP could significantly impact outcomes in SCI [2,10,17]. Increased ITP, resulting from spinal cord edema, can lead to decreased perfusion within the rigid confines of the spinal canal, echoing the pathophysiological processes encountered in compartment syndromes [12,13]. Timing of post-injury treatment is crucial for neuronal survival [30,68], and the detrimental “ischemia-edema-ischemia” cycle established by elevated ITP underscores the critical need for early intervention to disrupt this process. On a molecular level, managing ITP and enhancing SCPP not only increases the oxygen, nutrients, and neurotrophic factors supply, which directly influence neuroplasticity, but also can modulate the expression of genes associated with neuroregeneration, synaptic remodeling, and axonal growth [7,68,69]. Based on existing evidence, decompression surgery is most effective when performed within 24 h post-injury, as this time window is associated with significantly higher rates of ASIA grade improvement and functional recovery. Studies such as Zhu et al. [17] and Fehlings et al. [66] have emphasized that early surgical intervention not only mitigates secondary injury mechanisms but also enhances neuroplasticity through improved spinal cord perfusion pressure (SCPP). Future research should focus on refining protocols to ensure that patients receive decompression surgery within this optimal time frame.

Central to our discussion is the concept of neuroplasticity, which our findings suggest may be facilitated by reducing ITP and enhancing SCPP through surgical intervention [15,16,26]. By alleviating mechanical compression and optimizing the microenvironment for neural tissue, decompressive laminectomy and expansive duraplasty may set the stage for neuroplastic changes, offering patients a previously unimaginable potential for recovery [1,2,14].

The inclusion of both human and animal studies, while introducing translational heterogeneity, enabled a dual analysis of pathophysiological mechanisms and functional recovery. While animal models have elucidated key neuroplasticity mechanisms such as BDNF upregulation (+142% vs. sham, *p* < 0.001) and CSPG modulation, human valida-tion remains partial—limited to indirect evidence through CSF biomarkers (e.g., BDNF levels correlating with ASIA motor scores in Zhu et al. [17]) and postoperative functional improvements [25]. Moreover, limited human data revealed time-dependent clinical improvements, with early surgical intervention (<24 h post-injury) associated with a 3.1-fold higher likelihood of ASIA grade improvement (OR = 3.1). However, only 14/23 human studies (61%) reported standardized metrics like ASIA or SCIM-III, introducing potential selection bias. For instance, Werndle et al. documented SCPP optimization without correlating these findings to functional outcomes, highlighting a critical gap in translational reporting [14]. Critical translational gaps include the lack of human histopathological correlation, standardized injury models mirroring clinical biomechanics, and longitudinal biomarkers to track neuroplasticity across species. Future research should adopt unified protocols to harmonize molecular and clinical endpoints, particularly in studies evaluating ITP-SCPP dynamics.

The role of neuroplasticity in SCI recovery is increasingly recognized, with evidence pointing to molecular mechanisms triggering intracellular signaling pathways such as MAPK/ERK and TrKA/MAPK, which promote synaptogenesis and the reorganization of neural circuits [69,70]. The surgical intervention described in our study may facilitate these processes by altering the expression of molecules like BDNF and NGF, which are critical for supporting neuroplastic changes [7,68].

Our methodology for addressing increased ITP included both surgical and non-surgical strategies, with a focus on expansive duraplasty to relieve the pressure exerted by the swollen spinal cord against the dura mater [1,2,14]. The preference for expansive duraplasty over laminectomy alone is supported by emerging evidence that reducing ITP and increasing SCPP can directly influence neurological recovery [2,9,25].

While decompressive laminectomy and expansive duraplasty promote adaptive neuroplasticity and functional recovery, maladaptive neuroplasticity, such as neuropathic pain and spasticity, remains a significant challenge following spinal cord injury. Negative neuroplasticity arises from mechanisms like central sensitization and excitotoxicity, which can lead to hyperexcitability in neural circuits. Management strategies for these complications include pharmacological interventions, such as gabapentinoids (e.g., pregabalin) and GABAergic drugs (e.g., baclofen), which aim to restore inhibitory tone and reduce neuronal overactivity [71]. Additionally, non-pharmacological approaches, including physical therapy and neuromodulation techniques like transcutaneous electrical nerve stimulation (TENS), have shown promise in alleviating neuropathic pain and mitigating spasticity [72]. Future research should focus on integrating these strategies into postoperative care protocols to address the dual aspects of adaptive and maladaptive neuroplasticity effectively.

This review highlights the multifaceted benefits of decompressive laminectomy and duraplasty in SCI. By alleviating intrathecal pressure (ITP) and enhancing spinal cord perfusion pressure (SCPP), these procedures not only mitigate secondary injury but also foster an environment conducive to neuroplasticity [15,16,25]. Key molecular mechanisms include the modulation of neurotrophic factors such as brain-derived neurotrophic factor (BDNF) and nerve growth factor (NGF) [7,68] and the reduction in inhibitory molecules like chondroitin sulfate proteoglycans (CSPGs) [28,29].

### Limitations

While clinical and experimental data are promising, limitations exist, including small sample sizes and variability in surgical techniques [1,2,24]. Further research should focus on identifying optimal timing for intervention [30,68], exploring the long-term effects on maladaptive neuroplasticity [15,22], and investigating novel adjunctive therapies to enhance surgical outcomes [20,21,73].

The inclusion of both human and animal studies in this systematic review, while providing a comprehensive exploration of pathophysiological mechanisms and surgical outcomes, introduces inherent heterogeneity that may limit direct translational conclusions. This methodological approach was necessitated by the complementary strengths of each model: animal studies offer controlled environments to elucidate molecular pathways like BDNF upregulation and CSPG modulation, while human clinical data provide real-world insights into functional recovery and complications. However, the synthesis of these disparate populations risks conflating mechanistic findings from rodent models with clinical outcomes influenced by variables such as injury severity, comorbidities, and rehabilitation protocols. To mitigate this, we employed rigorous quality assessment tools (SYRCLE for animal studies, Cochrane for clinical trials) and analyzed molecular markers separately from functional outcomes. Future research should prioritize translational studies that bridge this gap through standardized injury models mirroring human biomechanics and longitudinal biomarkers tracking neuroplasticity across species. While heterogeneity remains a limitation, the combined analysis reveals conserved mechanisms—particularly ITP reduction and SCPP optimization—that warrant targeted investigation in both preclinical and clinical settings.

## 5. Conclusions

SCI remains a challenging neurological condition with limited therapeutic options for meaningful recovery. Our systematic review highlights the potential of decompressive laminectomy and expansive duraplasty as surgical interventions to alleviate ITP, enhance SCPP, and foster neuroplasticity. The findings suggest that these procedures mitigate secondary injury mechanisms and create a microenvironment conducive to neural repair through the modulation of key molecular pathways, including increased expression of neurotrophic factors like BDNF and NGF and reduced inhibitory CSPG levels.

Despite promising results, current evidence is constrained by small sample sizes, study design heterogeneity, and surgical technique variability. Further high-quality research, particularly well-designed randomized controlled trials, is needed to establish standardized protocols, optimize surgical timing, and explore synergistic therapies that may enhance neuroplasticity and functional recovery. Ultimately, a deeper understanding of how decompressive surgical strategies influence neuroregeneration could pave the way for improved clinical outcomes in SCI patients, offering renewed hope for functional restoration and enhanced quality of life.
neurolint-17-00057-t002_Table 2Table 2Summary of the included studies in the systematic review. Each study is categorized by its study design, population, intervention, and outcomes.
StudyStudy DesignPopulationInterventionOutcomes1Garg et al., 2022 [25]Clinical—Retrospective18 patients (SCI)Decompressive laminectomy + duraplastyImproved ITP, SCPP, and neuroplasticity markers. 1 AIS grade ↑SCIM-III: +16.52Phang et al., 2015 [13]Clinical—Observational25 patients (SCI)Perfusion monitoringImproved SCPP and pressure reactivity. WISCI II ≥ 5: 57% vs. 29% (control). No ASIA Score3Curt et al., 2008 [1]Clinical—ReviewVariable (SCI)NANeuroplasticity mechanisms4Kornblith et al., 2013 [2]Clinical—Multicenter150 patients (SCI)Mechanical ventilation strategiesImproved extubation rates5Lenehan et al., 2012 [3]Clinical—EpidemiologicalPopulation-basedNAEpidemiological insights6Thietje et al., 2011 [4]Clinical—Retrospective62 patients (Deceased SCI)Mortality analysisMortality and cause insights7Keefe et al., 2017 [5]Preclinical—AnimalRodent modelsNeurotrophic factor modulationIncreased BDNF, NGF levels8Stoyanova et al., 2021 [6]Preclinical—AnimalRodent modelsGhrelin-mediated plasticityEnhanced regeneration9Yue et al., 2020 [7]Clinical—Prospective35 patients (SCI)Perfusion protocolsEnhanced functional recovery10Saadoun et al., 2020 [8]Clinical—Observational20 patients (SCI)Targeted perfusion therapyReduced edema, improved outcomes11Leonard et al., 2015 [24]Preclinical—AnimalRodent modelsSubstance P modulationReduced inflammation and edema12Punjani et al., 2023 [15]Preclinical—ReviewMixed human/animal dataPlasticity pathwaysHighlighted neuroplasticity mechanisms13Zhu et al., 2019 [69]Clinical—Retrospective30 patients (SCI)Durotomy with duroplastyImproved motor function and reduced intrathecal pressure. ASIA Motor Score Δ: +14.4. Bladder control: 73%14Ahuja et al., 2017 [8]Clinical—Systematic ReviewVariable population (SCI)Repair and regeneration strategiesInsights on neuroplasticity and axonal repair15Leonard et al., 2013 [67]Preclinical—AnimalRodent modelsSubstance P modulationReduced inflammation and improved functional outcomes16Gotz et al., 2015 [19]Preclinical—AnimalRodent modelsAstrocytic plasticity interventionsEnhanced synaptic remodeling and axonal regeneration17Lau et al., 2011 [26]Preclinical—AnimalLamprey brain modelsNeurite sprouting post-SCIIncreased synapsin expression and sprouting18Anjum et al., 2020 [10]Clinical—Observational50 patients (SCI)Inflammation-targeted therapiesReduced secondary damage and improved recovery19Dimou and Gallo, 2015 [64]Preclinical—ReviewVarious animal modelsNG2-glia functionsInsights into glial plasticity and neurogenesis20Guo et al., 2019 [70]Preclinical—AnimalMouse modelsGene expression modulationIdentification of genes promoting regeneration21Cafferty et al., 2010 [74]Preclinical—AnimalRodent modelsGrowth-associated genesEnhanced axonal sprouting and plasticity22Cozzens et al., 2013 [27]Clinical—Systematic ReviewVariable population (SCI)Cervical spine and spinal cord injury managementGuidelines for early intervention23Xing et al., 2022 [51]Preclinical—AnimalRat modelsPI3K/AKT signaling pathwaysImproved axonal growth and synaptogenesis24Bobinger et al., 2018 [75]Preclinical—ReviewMixed modelsApoptotic pathways in neural injuryInsights on reducing cell death post-injury25Lee et al., 2010 [73]Preclinical—AnimalRodent modelsGhrelin for apoptosis inhibitionImproved functional recovery26Le Feber et al., 2016 [76]Preclinical—In vitroNeural culturesNeuronal damage progression in ischemiaModeling SCI-like ischemic conditions27Stoyanova et al., 2022 [77]Preclinical—AnimalRodent modelsHypoxia-induced Pax6 modulationEnhanced neuronal survival and regeneration28Galtrey and Fawcett, 2007 [23]Preclinical—ReviewMixed modelsRole of CSPGs in regenerationReduction in inhibitory signaling29Saadoun et al., 2019 [19]Clinical—Observational25 patients (SCI)Perfusion-targeted therapiesReduced edema and improved SCPP30Sun et al., 2018 [52]Preclinical—AnimalMouse modelsStem cells and exerciseEnhanced recovery via PI3K/AKT pathways31Grassner et al., 2018 [16]Clinical—ReviewVariable populationSpinal meninges in SCINeuroanatomical insights into recovery32Sharma et al., 2022 [37]Clinical—Retrospective20 patients (SCI)Magnetic resonance imaging in perfusion monitoringImproved spinal cord perfusion visualization33Miao et al., 2023 [78]Preclinical—AnimalRodent modelsNeuroplasticity via TrKA pathwaysEnhanced neurite elongation and recovery34Werndle et al., 2014 [14]Clinical—Observational30 patients (SCI)Perfusion pressure monitoringReduced secondary injury through SCPP improvements35Kwon et al., 2009 [12]Clinical—Randomized40 patients (SCI)Intrathecal pressure monitoringImproved outcomes via drainage protocols36Chen et al., 2012 [30]Preclinical—AnimalRat modelsBDNF signaling in synaptogenesisEnhanced recovery of motor function37Varsos et al., 2015 [11]Clinical—Observational30 patients (SCI)Spinal perfusion pressure dynamicsReduced pressure-related damage38Leonard et al., 2015 [68]Preclinical—AnimalRodent modelsEdema and hemorrhage contributionsReduction in post-injury complications39Fehlings et al., 2006 [66]Clinical—Systematic ReviewVariable population (SCI)Timing of interventionGuidelines for early surgical decompression40Hu et al., 2023 [32]Preclinical—AnimalRodent modelsMulti-molecular interactions post-SCIInsights on recovery mechanisms41Hill et al., 2019 [79]Preclinical—AnimalRodent modelsReactive astrocyte modulationImproved synaptic plasticity42Ahuja et al., 2017 [29]Clinical—Retrospective50 patients (SCI)Surgical repair strategiesImproved outcomes via axonal repair43Kheram et al., 2023 [80]Clinical—Observational11 patients (SCI)Perfusion-targeted interventionsImproved SCPP and reduced edema


## Figures and Tables

**Figure 1 neurolint-17-00057-f001:**
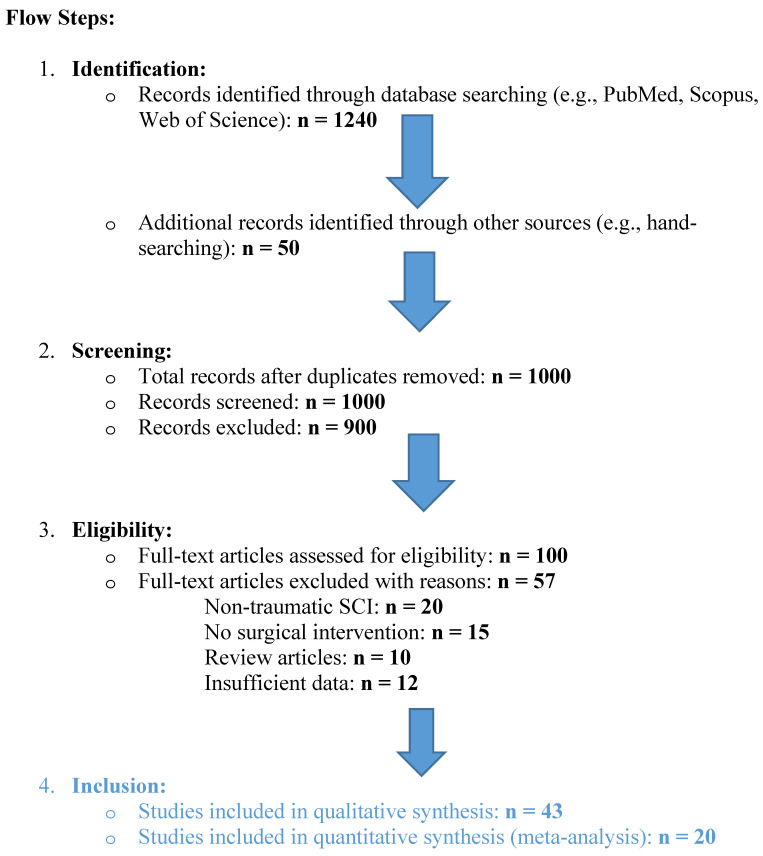
PRISMA Flow diagram for systematic review, flow steps.

**Figure 2 neurolint-17-00057-f002:**
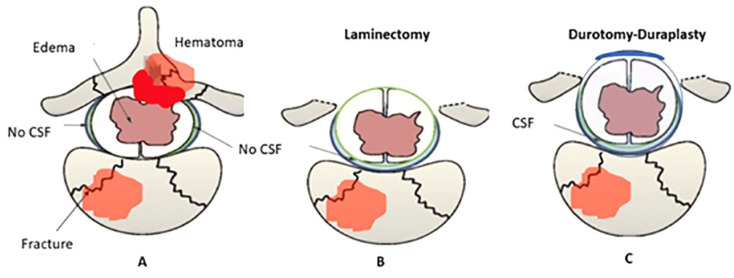
Schematic representation of spinal cord injury and surgical interventions. The progression from A to C demonstrates the surgical approach to severe spinal cord injury, aiming to restore anatomical integrity and promote neuroplasticity. (**A**) Initial condition post-trauma showing a vertebral fracture with adjacent hematoma (epidural compression ins highlighted in red color) and edema, leading to spinal cord compression. There is no cerebrospinal fluid (CSF) visible around the spinal cord due to the compression. (**B**) Post-decompressive laminectomy with removal of the posterior vertebral arch to relieve pressure on the spinal cord. The spinal cord shows restored space around it, but still no CSF is visible, indicating potential ongoing compression or adhesions. (**C**) After durotomy and duraplasty, CSF is visible around the spinal cord, indicating decompression. This procedure facilitates an environment for potential neuroplasticity and recovery of function.

**Figure 3 neurolint-17-00057-f003:**
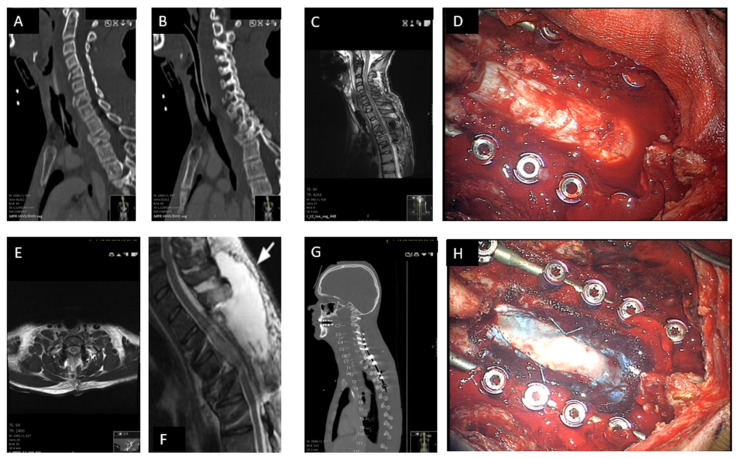
Case showing preoperative imaging, surgical interventions, and postoperative imaging in severe spinal cord injury management. (**A**) This panel demonstrates multiple unstable cervical and thoracic fractures, highlighting the severe structural compromise and the necessity for surgical intervention. (**B**) Damage to the posterior osteoligamentary structures as a result of injury is depicted, underscoring the extent of trauma and the resultant instability requiring stabilization. (**C**) A severe edema and catastrophic spinal cord contusion at levels C4-5 is shown, illustrating the urgency and complexity of the neurosurgical challenge. (**D**) The surgical field post-decompressive laminectomy is displayed, revealing the direct aftermath of removing the laminae to relieve pressure on the spinal cord. (**E**) The effective decompression post-surgery is evident, showcasing the alleviation of pressure and the potential for improved neurological function. (**F**) Laminectomy at levels C4-5-6 is shown, which is a surgical procedure to remove a portion of the lamina. The arrow shows the extended defect with seroma. (**G**) Cervicothoracic stabilization is displayed, indicating the surgical measures taken to secure the spinal integrity following decompression. (**H**) Duraplasty expansion is illustrated, which is used to provide additional space for the spinal cord and nerves by expanding the dural sac.

## Data Availability

The original contributions presented in the study are included in the article, further inquiries can be directed to the corresponding author.

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
