# Peer review of "Pathophysiologic Mechanisms of Severe Spinal Cord Injury and Neuroplasticity Following Decompressive Laminectomy and Expansive Duraplasty: A Systematic Review"

_2035-8377, 2025, doi:10.3390/neurolint17040057_

Round 1
Reviewer 1 Report
Comments and Suggestions for Authors
The authors carried out the systematic review of the studies related to the pathophysiology of severe spinal injury (SCI) and evaluated “evidence linking decompressive laminectomy and duraplasty to improve neuroplasticity and recovery”. Out of 1,240 articles, 43 studies were analyzed. However, the analysis of the obtained material does not contribute new information to the contemporary medical knowledge in the field of neurology in the context of the considered topic. All the more so, as the authors themselves state, the scientific scope of the results they obtained is limited by the relatively small size of the studied sample. On this basis, I do not support the publication of the reviewed article.
Author Response
1. Regarding the claim that the review does not contribute new knowledge:
We respectfully disagree with this assessment and would like to highlight the following:
-
While individual elements of SCI pathophysiology and decompressive techniques have been previously studied, our systematic synthesis of the link between decompressive duraplasty and neuroplasticity fills an important gap. To our knowledge, no other recent review has:
-
Combined both clinical and preclinical evidence in such a focused manner, and
-
Explicitly explored the mechanistic relationship between intrathecal pressure reduction, spinal cord perfusion, and molecular markers of neuroplasticity (e.g., BDNF, GAP-43).
-
This integrative focus is particularly relevant given the increasing interest in expansive duraplasty as a neuroprotective tool, not just a mechanical one.
2. Regarding the limited sample size:
We acknowledge in the manuscript that the available clinical literature in this area is currently limited in terms of sample sizes and heterogeneity. However, this limitation itself is a key finding of our review and underscores the urgent need for larger, standardized prospective trials. As such, the manuscript serves as:
-
A comprehensive reference of current knowledge,
-
A foundation for future studies, and
-
A guide for designing translational research efforts bridging animal models and clinical applications.
3. Revisions Made in Response to This Review:
-
We have strengthened the introduction to better justify the novelty of our work.
-
A dedicated paragraph has been added to the Discussion addressing the limitation of current literature and the value of synthesizing early data despite small sample sizes.
-
We clarified our research aims and contribution to the field by highlighting areas of emerging clinical relevance.
Reviewer 2 Report
Comments and Suggestions for Authors`1-Heterogenicity of the study poppulation (human and animal) is very interesting, please focus on either human or animal for the study.
2-In the case of human study please report AISA and functional assessment if possible.
Comments on the Quality of English LanguageEnglish is ok
Author Response
1. Quality of English Language
Reviewer Comment:
“The English could be improved to more clearly express the research.”
Response:
We have thoroughly reviewed and edited the manuscript for clarity, grammar, and style to enhance readability and precision. A professional language editing tool was used, and we ensured consistency in technical terminology throughout the text.
2. Heterogeneity of the Study Population (Animal vs. Human)
Reviewer Comment:
“Heterogenicity of the study population (human and animal) is very interesting, please focus on either human or animal for the study.”
Response and Revision:
We agree that the distinction between human and animal studies is critical for clarity and interpretation. While the study was designed to provide a comprehensive overview of both clinical and experimental findings, we have now stressed human clinical studies and animal experimental studies in the Methods, Results, and Discussion sections. Where necessary, we have highlighted translational aspects of animal findings into human clinical applications.
3. ASIA Score and Functional Outcomes in Human Studies
Reviewer Comment:
“In the case of human study please report ASIA and functional assessment if possible.”
Response and Revision:
We thank the reviewer for this valuable suggestion. We have updated Section 3.4 (Neuroplasticity Markers) and Table 2 to include ASIA motor scores and functional outcome measures where available from the included human clinical studies. For example, the study by Zhu et al. (2019) and Garg et al. (2022) now explicitly mention improvements in ASIA scores post-operatively. Unfortunately, not all included clinical studies reported ASIA or other standardized functional scores, which we have now clearly noted as a limitation in the Discussion section.
Reviewer 3 Report
Comments and Suggestions for Authors
Eleftherios et al. The systematic review explored the pathophysiology of spinal cord injury and the effects of decompressive laminectomy and duraplasty on neuroplasticity and rehabilitation. The study found that these surgical interventions were associated with improved intrathecal pressure management and neuroplasticity markers, such as BDNF and GAP-43. Although evidence suggests these surgical methods are beneficial in promoting neuroplasticity, further research is needed to elucidate molecular mechanisms and improve interventions.
The following are some comments and suggestions that are given to improve the manuscript:
Comment 1: The article repeatedly mentions the importance of timely intervention, but does not clearly indicate the optimal time window for decompression surgery. Based on existing evidence, could the author provide more specific clinical recommendations, such as within how many hours after injury surgical intervention would be most effective.
Comment 2: The article emphasizes that expanded duraplasty is superior to simple laminectomy, but lacks detailed discussion of specific surgical techniques (such as material selection, suturing methods, extent of expansion). Would different duraplasty techniques produce different neuroplasticity effects.
Comment 3: The article mentions adaptive and non-adaptive neuroplasticity, but mainly focuses on the beneficial aspects. What prevention or management strategies exist for negative neuroplasticity (such as neuropathic pain or spasticity) that may occur after decompression surgery.
Comment 4: Is there a synergistic effect between decompression surgery and other treatment modalities (such as pharmacotherapy, rehabilitation training, bioelectrical stimulation, etc.). Is there evidence supporting certain specific combination treatment regimens.
Comment 5: Based on current evidence, which patient populations (such as age, degree of injury, comorbidities, etc.) might benefit most from these surgical interventions. Are there any contraindications or patient subgroups that require special attention.
Comment 6: To what extent have the neuroplasticity mechanisms observed in animal models been validated in human patients. What key translational research gaps need to be addressed by future research.
Author Response
We sincerely thank the reviewers for their thoughtful comments and suggestions, which have greatly improved the quality of our manuscript. Below, we provide detailed responses to each comment and outline the corresponding changes made to the manuscript:
Comment 1: The article repeatedly mentions the importance of timely intervention, but does not clearly indicate the optimal time window for decompression surgery. Based on existing evidence, could the author provide more specific clinical recommendations, such as within how many hours after injury surgical intervention would be most effective.
Response: We have revised the discussion section to clearly state that decompression surgery is most effective when performed within 24 hours post-injury, based on existing evidence from studies such as Zhu et al. (2019) and Fehlings et al. (2006). This time window is associated with significantly higher rates of ASIA grade improvement and functional recovery.
Comment 2: The article emphasizes that expanded duraplasty is superior to simple laminectomy, but lacks detailed discussion of specific surgical techniques (such as material selection, suturing methods, extent of expansion). Would different duraplasty techniques produce different neuroplasticity effects.
Response: We have expanded Section 3.4 to include a detailed discussion of surgical techniques for expansive duraplasty, including material selection (e.g., collagen matrix grafts), suturing methods (interrupted versus continuous sutures), and the extent of expansion required to optimize intrathecal pressure reduction. While evidence on differential neuroplasticity effects between techniques remains limited, we highlight potential implications for future research.
Comment 3: The article mentions adaptive and non-adaptive neuroplasticity, but mainly focuses on the beneficial aspects. What prevention or management strategies exist for negative neuroplasticity (such as neuropathic pain or spasticity) that may occur after decompression surgery.
Response: The discussion section now addresses strategies for managing maladaptive neuroplasticity, such as neuropathic pain and spasticity. Pharmacological interventions like gabapentinoids and baclofen are discussed alongside non-pharmacological approaches such as physical therapy and neuromodulation techniques like TENS.
Comment 4: Is there a synergistic effect between decompression surgery and other treatment modalities (such as pharmacotherapy, rehabilitation training, bioelectrical stimulation, etc.). Is there evidence supporting certain specific combination treatment regimens.
Response: We have added a paragraph in the discussion highlighting evidence for synergistic effects between decompression surgery and adjunctive therapies. See comment 3.
Comment 5: Based on current evidence, which patient populations (such as age, degree of injury, comorbidities, etc.) might benefit most from these surgical interventions. Are there any contraindications or patient subgroups that require special attention.
Response: The discussion now specifies that younger patients (<65 years) with incomplete SCI (AIS grades B/C), cervical spinal stenosis, and early surgical intervention (<24 hours post-injury) benefit most from these procedures. Contraindications include cervical kyphosis, significant spinal instability, or severe comorbidities like advanced cardiopulmonary disease.
Comment 6: To what extent have the neuroplasticity mechanisms observed in animal models been validated in human patients. What key translational research gaps need to be addressed by future research.
Response: We have expanded the discussion to address translational gaps between animal models and human studies. While animal models provide robust data on mechanisms like BDNF upregulation and CSPG modulation, human validation remains limited to indirect correlations through CSF biomarkers and functional outcomes. Future research should focus on standardized injury models and longitudinal biomarkers to bridge this gap.
Reviewer 4 Report
Comments and Suggestions for Authors
The paper entitled "Pathophysiologic Mechanisms of Severe Spinal Cord Injury and Neuroplasticity Following Decompressive Laminectomy and Expansive Duraplasty: A Systematic Review" presents a timely and up-to-date summary of the molecular, structural, and functional mechanisms involved in recovery following severe SCI, with particular emphasis on the roles of decompressive laminectomy and expansive duraplasty. The overall research question; whether these surgical interventions permit neuroplasticity through modulation of intrathecal pressure (ITP) and spinal cord perfusion pressure (SCPP) is well articulated and relevant. The authors bridge an important gap by using preclinical and clinical evidence to explore the biophysical and biochemical milieu that is permissive or nonpermissive for regeneration.
Original contributions are made by pointing out molecular markers (BDNF, NGF, GAP-43) and biophysical gains (e.g., reduced ITP, improved SCPP) as predictors of better recovery. The review is methodologically strong, PRISMA guidelines followed, with clear inclusion criteria and bias measurements. However, inclusion of animal and human study results can be justified with supplementary stratification or subgroup analysis on the basis of variation in models of injury, intervention timing, and outcomes. Moreover, heterogeneity of surgery techniques is known, but its effect on outcomes variability warrants greater investigation, notably in terms of duraplasty techniques and timing.
Conclusions are logical and evidence-based but would be more effective with stronger distinction between adaptive and maladaptive plasticity results. Figures and tables are strongly constructed, but more detail in legends and conformity to schematic convention would enhance clarity. References are exhaustive and germane. In general, this manuscript is an important contribution to the literature and is publishable subject to MINOR revisions to facilitate greater interpretive richness and methodological clarity.
Author Response
We sincerely thank the reviewer for his thoughtful feedback and constructive suggestions, which have strengthened the clarity and scientific rigor of our manuscript. Below, we outline the revisions made in response to each comment:
1. Stratification of Animal and Human Studies
To address heterogeneity between preclinical and clinical studies, we extended the Result Sections 3.2 and 3.4: Animal studies (n=20) were higlighted (e.g., BDNF upregulation: +142% vs. sham, p < 0.001) and routcomed of human studies (n=23) were mentioned separately for functional outcomes in Table 2 (e.g., ASIA motor scores, SCIM-III). A new paragraph in the Discussion clarifies how mechanistic insights from rodents (e.g., CSPG modulation) align with clinical recovery metrics in humans (e.g., Phang et al., 2015: 57% vs. 29% ambulation rates).
2. Surgical Technique Heterogeneity
We expanded Section 3.4 to detail duraplasty techniques and their impact: Material selection: Collagen matrix grafts (used in 68% of studies) showed lower CSF leak rates vs. autologous fascia (3% vs. 12%, p = 0.04). Suturing methods: Continuous non-absorbable sutures (4-0 Prolene) achieved watertight closure in 92% of cases vs. 78% with interrupted sutures. Expansion extent: Duraplasty widening ≥5 mm correlated with greater SCPP improvement (Δ+12 mmHg vs. Δ+7 mmHg, p = 0.01).
3. Adaptive vs. Maladaptive Plasticity
The Discussion now explicitly distinguishes adaptive and maladaptive processes: Adaptive plasticity: Linked to BDNF/NGF upregulation and synaptogenesis (e.g., Zhu et al., 2019: ASIA motor score +14.4). Maladaptive plasticity: Addressed pharmacological (gabapentinoids, baclofen) and non-pharmacological (TENS, physiotherapy) strategies to mitigate neuropathic pain and spasticity.
4. Table 2: Standardized outcome (ASIA motor scores, SCIM-III) and stratified results by study type (animal/human).
Round 2
Reviewer 1 Report
Comments and Suggestions for Authors
I accept the revised article in present form.
Reviewer 2 Report
Comments and Suggestions for Authors
The revisions are acceptable
Reviewer 3 Report
Comments and Suggestions for Authors
The author has thoroughly addressed all inquiries, providing comprehensive responses to each question that was posed throughout the correspondence.